# SURGICAL PREDICTION WITH INTERPRETABLE LATENT REPRESENTATION

## ABSTRACT

Given the risks and cost of surgeries, there has been significant interest in exploiting predictive models to improve perioperative care. However, due to the high dimensionality and noisiness of perioperative data, it is challenging to develop accurate, robust and interpretable encoding for surgical applications. We propose *surgical VAE (sVAE)*, a representation learning framework for perioperative data based on variational autoencoder (VAE). sVAE provides a holistic approach combining two salient features tailored for surgical applications. To overcome performance limitations of traditional VAE, it is *prediction-guided* with explicit expression of predicted outcome in the latent representation. Furthermore, it *disentangles* the latent space so that it can be interpreted in a clinically meaningful fashion. We apply sVAE to two real-world perioperative datasets and the open MIMIC-III dataset to evaluate its efficacy and performance in predicting diverse outcomes including surgery duration, postoperative complication, ICU duration, and mortality. Our results show that the latent representation provided by sVAE leads to superior performance in classification, regression and multi-task predictions. We further demonstrate the interpretability of the disentangled representation and its capability to capture intrinsic characteristics of surgical patients. While this work is motivated by and evaluated in the context of clinical applications, the proposed approach may be generalized for other fields using high-dimensional and noisy data and valuing interpretable representations.

## 1 INTRODUCTION

Hospitals have shown significant interest in exploiting machine learning to predict surgery-related outcomes from postoperative complications (Xue et al., 2021) to surgery duration (Jiao et al., 2020). More than 10% of surgical patients experience major postoperative complications, leading to increased mortality risk and need for a higher level of care (Tevis & Kennedy, 2013). Early identification of risk factors can be crucial to early intervention and improved outcomes (Kable et al., 2002). Perioperative care is also a major contributor to overall hospitalization expenses (Childers & Maggard-Gibbons, 2018). Predictions of surgery-related outcomes provides clinicians lead time in resource planning to reduce the cost of perioperative care (Denton et al., 2007) .

However, building a predictive model for perioperative data can be challenging due to the high dimensionality and complexity of perioperative data (Li et al., 2020). There are hundreds of perioperative variables of heterogenous types and the missing rates vary drastically across variables, leading to significant deterioration and instability in prediction performance (LE et al., 2016). Furthermore, the predictive models need to be interpretable from a clinical perspective for their acceptance and adoption in perioperative care.

Deep generative models based on variational auto-encoder (VAE) represent a promising approach to deal with the high dimensionality and complexity of perioperative data. Such models can extract the hidden nonlinear relationship of input variables while reducing the input dimensionality through latent representations (Gao et al., 2016; Kingma & Welling, 2013; Rezende & Mohamed, 2015). However, traditional VAEs have two major limitations for surgical predictions. First, as VAE training is traditionally unsupervised, predictive models based on VAE-encoded latent variables can suffer *performance degradation* (Feng et al., 2020; Locatello et al., 2019). Furthermore, the latent space from vanilla VAE models are often *entangled* (Burgess et al., 2018; Chen et al., 2018), which

makes it difficult for clinicians to interpret the nonlinear encoding and trust the predictions. Recent advances in VAEs aim to overcome the limitations separately. To improve predictive performance, semi-supervised VAEs are developed to integrate prediction loss into the training of latent encoding (Pálsson et al., 2019; Cheung et al., 2014; Wu et al., 2019), but such method does not guarantee disentanglement on latent dimensions other than isolating the specific dimension for prediction. On the other hand, several techniques have been proposed to disentangle the latent space (Chen et al., 2018; Kim & Mnih, 2018; Achille & Soatto, 2018; Burgess et al., 2018), but they are not concerned about optimizing downstream predictions. In fact, existing literature suggests that these disentanglement efforts have little help to predictive tasks (Feng et al., 2020; Locatello et al., 2019).

To meet the demand for high prediction performance and interpretable models in perioperative care, we propose *surgical VAE (sVAE)*, a representation learning framework specifically designed for high-dimensional and complex clinical data. sVAE provides a holistic framework that integrates the respective strengths of regularization-based disentanglement (Burgess et al., 2018; Chen et al., 2018) and guided encoding (Zhao et al., 2019; Feng et al., 2020; Zhou & Wei, 2020). It enhances predictive performance by organizing the latent encoding with respect to the predictive tasks explicitly. Furthermore, it groups patients into interpretable phenotypes in the latent space by disentangling the latent representation. We apply sVAE to two real-world perioperative datasets collected from 12,904 and 77,169 surgeries, respectively. To examine its generalizability to a broader setting, we applied sVAE to the open MIMIC-III clinical dataset and observed similar improvements. Our experiments demonstrate that sVAE consistently delivers superior predictive performance for classification and regression tasks while constructing interpretable latent embedding that captures the intrinsic information of original clinical data and achieves better recovery of original data.

While this work is motivated by and evaluated in the context of clinical applications, we note that the proposed approach may be generalized for other fields that demand accurate predictions and interpretable representations based on high-dimensional and noisy data.

## 2 RELATED WORKS

As a representation learning framework, traditional VAE has been applied to different clinical applications such as abnormality detection (Baur et al., 2018) or clustering tasks (Zhao et al., 2019). Recent research advances VAE in separate directions including semi-supervised models integrating prediction tasks and disentanglement of unsupervised models.

**Integration of prediction tasks.** These exists various types of semi-supervised VAEs that integrate prediction task as a regularizer in latent encoding. Such models, including M2 (Kingma et al., 2014), pi-VAE (Zhou & Wei, 2020), i-VAE (Sorrenson et al., 2020) and others (Wu et al., 2019; Pálsson et al., 2019) organize the latent distribution with respect to the prediction (or conditioning) variable, hence the prediction information is contained in the representation in latent space. Intuitively the integration of prediction label will lead to easier downstream prediction tasks, as latent representations should have been well-separated by the task. However, as discussed in recent publications (Feng et al., 2020; Locatello et al., 2019), a well-trained VAE model does not benefit downstream prediction, often referred to as "*Good ELBO, Bad Inference*". The extended semi-supervised models adds an explicit loss term to address this issue (Zhao et al., 2019; Feng et al., 2020) but there are still remaining issues. On one hand, if we assume all latent dimensions to be dependent on the prediction label ( (Zhao et al., 2019)), then as Sorrenson et al. (2020) point out, the disentangling effects only work when the prediction (conditioning) label acts as the cause for the rest latent variables, which obviously does not hold in clinical applications. On the other hand, if we assume the predicted label is independent from the rest of latent dimensions (Hejna et al.; Feng et al., 2020), this assumption is usually violated in model training without a designated constraint. Hejna et al. propose a factor prediction method that emphasizes the independence of prediction dimension from other latent dimensions. However, the encoded position on the prediction dimension is not guided to convey any specific meanings, and the rest of latent dimensions are not disentangled from each other. In contrast, our proposed sVAE maximizes the disentanglement between all latent dimensions and organizes all samples on a prediction dimension based on the predicted label. By doing so, it naturally distills the predictive information to the single prediction dimension and the encoded coordinate can be directly interpreted as predicted values.

**Unsupervised disentanglement.** In $\beta$-VAE (Burgess et al., 2018), the disentanglement is achieved by directly penalizing KL divergence. In subsequent studies, decomposing this KL term reveals that the total correlation (TC) term plays the essential role of disentanglement (Kim & Mnih, 2018; Chen et al., 2018; Achille & Soatto, 2018). As computing TC is intractable, several estimation metrics were proposed (Kim & Mnih, 2018; Chen et al., 2018), but they underestimate the true TC and lead to unbounded, inaccurate estimation (Locatello et al., 2019). In sVAE, we address this issue by evaluating the correlation in the sampled representation and minimizing a bounded metric.

Compared to these existing works, sVAE provides a holistic framework combining features of both disentanglement and predictions, enabling the possibility of phenotyping and explicit prediction, thereby meeting the demand for predictive performance and interpretability of surgical models. With more stable estimation of correlation, we hope to achieve both better disentanglement and meanwhile more accurate representation of samples.

## 3 BACKGROUND

**Notations:** We denote $\boldsymbol{X} \in \mathbb{R}^{n \times p}$ as the numerical embedding of the perioperative EHR data, where clinical texts, string categories, and time series have been represented by numerical vectors as illustrated in Appendix Fig. A1. $y \in \mathbb{R}$ represents the prediction tasks, either discrete (classification prediction) or continuous (regression prediction), and is only available after the surgery.

A vanilla VAE consists of an encoder and a decoder, linked by a loss function. The encoder is usually represented by a neural network with parameters $\phi$, which outputs $q_\phi(\boldsymbol{z}|\boldsymbol{x})$, a posterior Gaussian probability density of latent variable $\boldsymbol{z} \in \mathbb{R}^d$. The decoder is another neural network, denoted by $p_{\boldsymbol{\theta}}(\boldsymbol{x}|\boldsymbol{z})$. Its input is the latent representation $\boldsymbol{z}$ and outputs the probability distribution of the data. As VAE attempts to both reconstruct data and enforce smooth latent distribution, the objective function, usually referred to as Evidence Lower BOund (ELBO), is designed as follows:

$$l(\boldsymbol{\phi}, \boldsymbol{\theta}) = \sum_{i=1}^{N} E_{\boldsymbol{z} \sim q_\phi(\boldsymbol{z}|\boldsymbol{x}^{(i)}])} [\log p_{\boldsymbol{\theta}}(\boldsymbol{x}^{(i)}|\boldsymbol{z})] - KL(q_\phi(\boldsymbol{z}|\boldsymbol{x})||p(\boldsymbol{z})) \tag{1}$$

The regularizer measures the Kullback-Leibler divergence between the encoded data's distribution and the assumed distribution of $\boldsymbol{z}$. This regularizer term keeps the latent space sufficiently diverse and meanwhile organize all latent representations conforming to the prior belief.

### 3.1 INTEGRATION OF TASK VARIABLES

Recent development developed various variants of VAE to incorporate prediction label into the model training. In pi-VAE and similar models (Zhou & Wei, 2020; Khemakhem et al., 2020; Sorrenson et al., 2020), prediction label is used for conditional distribution to obtain improved disentanglement of latent dimensions. The loss function is defined as:

$$l(\boldsymbol{\phi}, \boldsymbol{\theta}) = \sum_{i=1}^{N} [-E_{\boldsymbol{z} \sim q_\phi(\boldsymbol{z}|\boldsymbol{x}^{(i)}, u_i)} [\log p_{\boldsymbol{\theta}}(\boldsymbol{x}^{(i)}|\boldsymbol{z}, u_i)] + KL(q_\phi(\boldsymbol{z}|\boldsymbol{x}, u_i)||p(\boldsymbol{z}|u_i))] \tag{2}$$

where $u$ is another observed variable (or prediction label $y$ in our case).

A different approach of integrating $y$ is to let $y$ take one of the latent dimensions (Zhao et al., 2019; Hejna et al.; Feng et al., 2020). Depending on the assumptions, the expression of loss function can be different. We refer such approach as smooth-ELBO (Feng et al., 2020), defined by:

$$l(\boldsymbol{\phi}, \boldsymbol{\theta}) = \sum_{i=1}^{N} [-E_{q_\phi(\boldsymbol{z}|\boldsymbol{x}^{(i)})} [\log p_{\boldsymbol{\theta}}(\boldsymbol{x}^{(i)}|\boldsymbol{z})] + KL(f(y^{(i)})||q_\phi(y|\boldsymbol{x}^{(i)})) ] + KL(q_\phi(\boldsymbol{z}, y|\boldsymbol{x})||p(\boldsymbol{z}, y))$$

$$\tag{3}$$

where $f(\cdot)$ is a smoothing function when $y$ is discrete.

### 3.2 DISENTANGLEMENT

VAE models are usually highly entangled, making latent space hard to interpret (Chen et al., 2018; Burgess et al., 2018; Kim & Mnih, 2018). A simple modification by optimizing a heavily penalized

KL divergence achieves a high degree of disentanglement in image datasets (Burgess et al., 2018). The extra scalar parameter on the KL divergence is $\beta$ hence this variant is referred as $\beta$-VAE. It is not made explicit why penalizing $KL(q_\phi(\boldsymbol{z}|\boldsymbol{x}^{(i)})||p(\boldsymbol{z}))$ can lead to disentangled transformations, until the ablation study on the decomposition of the KL divergence showed that the total correlation (TC) term: $TC := KL(q_\phi(\boldsymbol{z})||\prod_{j=1}^{d} q_\phi(z_j))$ is the key factor. By emphasizing on the TC term with parameter $\beta$ in the loss function, we get $\beta$-TCVAE and FactorVAE (Chen et al., 2018; Kim & Mnih, 2018):

$$l(\boldsymbol{\phi}, \boldsymbol{\theta}, \beta) = \sum_{i=1}^{N} -E_{q_\phi(\boldsymbol{z}|\boldsymbol{x}^{(i)})}[\log p_{\boldsymbol{\theta}}(\boldsymbol{x}^{(i)}|\boldsymbol{z})] + KL(q_\phi(\boldsymbol{z}|\boldsymbol{x})||p(\boldsymbol{z})) + \beta * TC(\boldsymbol{z}) \quad (4)$$

where $z_j$ is the latent representation on dimension $j$. As TC($\boldsymbol{z}$) is intractable, several estimation metrics were proposed (e.g., density ratio (Kim & Mnih, 2018), batch sampling (Chen et al., 2018)).

## 4  sVAE

The aforementioned works have significant implication for surgical predictions. Conceptually, disentanglement helps reorganize the information in $\boldsymbol{X}$ into clinical characteristics/phenotypes that we may be able to discover; and the integration of task variables enable VAE to stratify surgical cases by predicted labels/values that we are interested in. Hence with such goals, we want our sVAE to 1) express explicit predictions on a latent dimension (serving as a predictor); 2) minimize the correlation between latent dimensions so that the latent dimension demonstrates interpretable phenotype. If we have an accurate explicit prediction as in 1), we would subsequently hope that it also has an accurate representation for other phenotypes, hence reconstructs the samples robustly.

**Distillation of Prediction.** Now consider a single case $\boldsymbol{x} \in \boldsymbol{X}$: as the prediction task $y$ is dependent on $\boldsymbol{x}$ (otherwise the problem is not meaningful), it is also dependent on its latent encoding $\boldsymbol{z}$. This fact can be represented as: $p(y, \boldsymbol{x}, \boldsymbol{z}) = p(y|\boldsymbol{z}, \boldsymbol{x})p(\boldsymbol{z}|\boldsymbol{x})p(\boldsymbol{x})$. To distill the prediction information $y$ from $\boldsymbol{z}$, let $g(\cdot)$ be a disentangling transformation function such that $g(\boldsymbol{z}) = [y, \boldsymbol{z}']$, where $\boldsymbol{z}'$ is a vector of the remaining information in $\boldsymbol{z}$ that is irrelevant to $y$. As $g(\cdot)$ is a disentangling function, all dimensions in $\boldsymbol{z}'$ are uncorrelated with each other too. This is similar to an independent component decomposition except that we explicitly isolate $y$ from other dimensions. Let $\boldsymbol{v} = [y, \boldsymbol{z}']$, then $p(\boldsymbol{v}) = p(y)p(\boldsymbol{z}')$. Since $\boldsymbol{v}$ is a transformation of $\boldsymbol{z}$, it is equivalently another latent representation of $\boldsymbol{x}$. Assuming $g(\cdot)$ is bijective, then $p(y, \boldsymbol{x}, \boldsymbol{z})$ can be re-written as $p(y, \boldsymbol{x}, \boldsymbol{z}) = p(y, \boldsymbol{x}, \boldsymbol{v}) = p(\boldsymbol{v}|y, \boldsymbol{x})p(y|\boldsymbol{x})p(\boldsymbol{x}) = p(\boldsymbol{z}'|\boldsymbol{x})p(y|\boldsymbol{x})p(\boldsymbol{x})$.

We introduce $g(\boldsymbol{z})$ for two purposes: it both forces the disentanglement in all latent dimensions for potential interpretability, and distills the predictive information from $\boldsymbol{z}$ and explicitly expresses it in a designated dimension. We use a neural network to implement $g(\boldsymbol{z})$, then its generative version $g(q_\phi(\boldsymbol{z}|\boldsymbol{x}^{(i)}))$ is equivalently another encoder. Let $g(q_\phi(\boldsymbol{z}|\boldsymbol{x}^{(i)})) = q_{\phi'}(\boldsymbol{v}|\boldsymbol{x}^{(i)})$, now we leverage the idea in Eq.3 for the learning of $p(\boldsymbol{z}'|\boldsymbol{x})$ and $p(y|\boldsymbol{x})$, and the TC term in Eq.5 for our assumption of disentanglement to hold true:

$$l(\boldsymbol{\phi}', \boldsymbol{\theta}, \alpha, \beta) = \sum_{i=1}^{N} [-E_{\boldsymbol{v} \sim q_{\phi'}(\boldsymbol{v}|\boldsymbol{x}^{(i)})}[\log p_{\boldsymbol{\theta}}(\boldsymbol{x}|\boldsymbol{v})] + \alpha * KL(f(y^{(i)})||q_{\phi'}(y|\boldsymbol{x}^{(i)})) \quad (5)$$
$$+ KL(q_{\phi'}(\boldsymbol{v}|\boldsymbol{x})||p(\boldsymbol{v})) + \beta * TC(\boldsymbol{v})]$$

where $\alpha$ and $\beta$ are the scalar coefficients for prediction loss and TC loss. Note that the smoothing function $f(y^{(i)})$ is unnecessary, especially when $y$ is a regressional task. In our experiments, we replace it by the posterior Gaussian likelihood $\log q_{\phi'}(y^{*(i)}|\boldsymbol{x}^{(i)})$ where $y^{*(i)}$ is the ground truth label, and set hyperparameters $\alpha = 1$ and $\beta = 10$. Compared to other prediction-guided models, the introduction of TC term not only guarantees the conformation to our independence assumption, but also eliminates the use of an additional predictor as prediction is explicitly expressed on a single dimension.

**Estimation of Correlation**. Recent studies show that the estimation of TC is usually misleading. As TC is measured by a KL-divergence from batch data $\boldsymbol{B}$, $TC = KL(q_\phi(\boldsymbol{z})||\prod_{j=1}^{d} q_\phi(z_j)) \approx \int_{\boldsymbol{z}} q_{\phi(\boldsymbol{B})}(\boldsymbol{z}) \log \frac{q_{\phi(\boldsymbol{B})}(\boldsymbol{z})}{\prod_{j=1}^{d} q_{\phi(\boldsymbol{B})}(z_j)}$. When $d$ is small, such approximation is appropriate. When $d$ is

large, there are several issues. First, when sampling $z \sim q_\phi(z)$, due to the curse of dimensionality, the input sparsity leads the estimation $q_{\phi(B)}(z)$ effectively 0, hence the estimation of such KL-divergence is insignificantly different from 0 and will not disentangle the latent space anymore (Kim & Mnih, 2018). In real applications, we may take the log term of the estimated KL divergence, and this inappropriate estimation would impose a negatively large, unbounded fluctuation in the loss minimization. Second, the estimation of TC using such method with mean representation is not consistent with the sampled representation (Locatello et al., 2019). Although several modified estimation methods were proposed (Kim & Mnih, 2018; Chen et al., 2018), with a higher dimensionality, these are no longer accurate and further lead to unintended shutdown of latent dimensions (Cheng et al., 2020). Besides, with more training iterations, the posterior variance of $q_{\phi'}(v|x)$ is usually smaller, amplifying such negative effects (see more results in Appendix A6).

To tackle this issue, we propose a simple estimation of disentanglement using singular value decomposition (SVD). The intuition behind our method is as follows: if the latent dimensions are disentangled, a linear transformation to a new coordinate system will not increase the variance on any new coordinates. In fact, it has been observed in medical imaging data that PCA/SVD-based metrics are equivalently effective as TC-based metrics in quantifying similarities (Guyader et al., 2016), which inspires the substitution of the inaccurate TC estimation by SVD. The evaluation of our proposed metric consists of three steps:

1. Draw Monte Carlo sampling from the latent distribution of current batch data (batchsize $:= B > d$). Concatenate into a $B \times d$ matrix $A = [\mathbf{v_1}, \mathbf{v_2}, ..., \mathbf{v_B}]^{\mathbf{T}}$. As each latent dimension may have different mean/variance, each dimension should be standardized.

2. Perform SVD to get the largest singular value $\sigma_1$, where $\sigma_1 \geq \sigma_2 \geq ... \geq \sigma_d \geq 0$.

3. Calculate $TC(v) = \frac{\sigma_1^2 - 1}{d - 1}$ (similar to the variance explained by the first principal component).

A main advantage of this correlation estimation method is the lack of hyperparameters or expensive computation, which should provide more stable training in an efficient way. Besides, our proposed evaluation strategy has 3 key properties:

1. $TC(v)$ is bounded by $[0, 1]$;

2. $TC(v)$ is minimized iff the all latent dimensions are uncorrelated (orthogonal);

3. $TC(v) = TC(v^{(1)}, v^{(2)}, ..., v^{(B)})$ estimated from a single batch data is the lower bound of the $TC$ of the whole latent space, calculated by $TC(v^{(1)}, v^{(2)}, ..., v^{(N)})$. Meanwhile, the minimization across all batches minimizes the upper bound of the whole latent space.

The detailed proofs are discussed in Appendix A1.

## 5 EXPERIMENTS

We evaluate sVAE on the open MIMIC-III dataset and two real-world perioperative datasets in terms of its perdictive performance and its efficacy in disentangling latent space and representing the true information. We further show how disentanglement enhances the interpretability of latent encoding from a clinical perspective.

### 5.1 DATASETS

**Delirium Data**: The first dataset was acquired from the electronic anesthesia records of all adult patients undergoing surgery at a large academic medical center from June 1, 2012 to August 31, 2016. Input variables include demographics, history of comorbidities, lab tests, medications, and statistical features extracted from time series. The prediction outcome is binary, indicating whether delirium was developed after surgery. In total, there are 12,904 patients with 52.65% of them developed postoperative delirium. This dataset is referred hereafter as *Delirium Data*.

**OR Data**: The second dataset was acquired from a large healthcare system spanning across eight hospitals, academic and community centers and surgical specialties from March 1, 2019 to October 31 2019. The input variables include the scheduled surgery duration, clinical notes describing

Table 1: Upper: Classification performance for MIMIC-III mortality prediction. Lower: Regression performance for MIMIC-III ICU LoS. Mean and standard error are reported after 5 random shuffles.

| Transformation Method (d=10) | Direct Prediction | | LR | | XGBoost | | SVM | | DNN | |
|---|---|---|---|---|---|---|---|---|---|---|
| | ROC AUC | Average Precision | ROC AUC | Average Precision | ROC AUC | Average Precision | ROC AUC | Average Precision | ROC AUC | Average Precision |
| PCA | - | - | .7630(.0047) | .6299 (.0084) | .7556 (.0038) | .6039 (.0102) | .7590 (.0079) | .6207 (.0117) | .7721 (.0048) | .6418 (.0099) |
| ICA | - | - | .6085(.0502) | .4924 (.0374) | .7231 (.0221) | .5798 (.0352) | .6488 (.0653) | .5173 (.0755) | .6146 (.0469) | .4962 (.0300) |
| AE | - | - | .7116(.0160) | .5646 (.0167) | .7043 (.0171) | .5513 (.0127) | .7016 (.0203) | .5594 (.0168) | .6861 (.0358) | .5515 (.0197) |
| VAE | - | - | .7004(.0308) | .5593 (.0336) | .7251 (.0213) | .5856 (.0324) | .6901 (.0355) | .5484 (.0420) | .6881 (.0407) | .5460(.0459) |
| $\beta$-TCVAE | - | - | .7133 (.0168) | .5728 (.0216) | .7333 (.0087) | .5962 (.0122) | .7001 (.0135) | .5606 (.0197) | .7017 (.0141) | .5630 (.0183) |
| pi-VAE | - | - | .7150 (.0266) | .5718 (.0263) | .6983 (.0146) | .5433 (.0132) | .7252 (.0192) | .5814 (.0241) | .6919 (.0353) | .5550 (.0292) |
| smooth-VAE | .8335 (.0026) | .7291 (.0048) | .8339 (.0024) | .7301 (.0000) | .8331 (.0028) | .7178 (.0047) | .8339 (.0025) | .7293 (.0041) | .8337 (.0025) | .7300 (.0051) |
| sVAE | **.8358 (.0028)** | **.7321 (.0048)** | **.8367 (.0021)** | **.7333(.0048)** | **.8358 (.0026)** | **.7211(.0042)** | **.8363 (.0022)** | **.7333(.0048)** | **.8367 (.0021)** | **.7332(.0050)** |
| Raw Data (d=38) | - | - | .7959(.0038) | .6759 (.0059) | .8248 (.0024) | .7214 (.0041) | .8022 (.0000) | .6812 (.0000) | .8358 (.0029) | .7323 (.0036) |

| Transformation Method (d=10) | Direct Prediction | | LR | | XGBoost | | SVM | | DNN | |
|---|---|---|---|---|---|---|---|---|---|---|
| | rMSE | $R^2$ | rMSE | $R^2$ | rMSE | $R^2$ | rMSE | $R^2$ | rMSE | $R^2$ |
| PCA | - | - | 9.148 (.2674) | .1919 (.0474) | 7.480 (.2333) | .4594 (.0338) | 7.382 (.2869) | .4732 (.0410) | 7.536 (.2136) | .4514 (.0311) |
| ICA | - | - | 9.637 (.3100) | .1022 (.0586) | 8.129 (.5795) | .3583 (.0956) | 10.59 (.2756) | -.0840 (.0563) | 10.27 (.2684) | -.0205 (.0534) |
| AE | - | - | 8.145 (.2510) | .3591 (.0393) | 7.795 (.1874) | .4131 (.0282) | 8.067 (.1331) | .3721 (.0207) | 8.285 (.1237) | .3377 (.1990) |
| VAE | - | - | 9.015 (.1910) | .2146 (.0333) | 7.685 (.2060) | .4295 (.0304) | 10.07 (.2084) | .0216 (.0407) | 9.256 (.3065) | .1715 (.0553) |
| $\beta$-TCVAE | - | - | 9.101 (.3343) | .1993 (.0581) | 7.803 (.2793) | .4121 (.0418) | 10.12 (.2763) | .0108 (.0532) | 9.420 (.3210) | .1422 (.0593) |
| pi-VAE | - | - | 8.992 (.2391) | .2190 (.0419) | 8.160 (.4325) | .3555 (.0676) | 8.953 (1.016) | .2157 (.1713) | 9.085 (1.024) | .1934 (.1751) |
| smooth-VAE | 6.710 (.2363) | .5648 (.0307) | 6.666 (.2442) | .5705 (.0314) | 6.639 (.2261) | .5741 (.0290) | 6.763 (.2531) | .5578 (.0330) | 6.724 (.2546) | .5629 (.0330) |
| sVAE | **6.590 (.2425)** | **.5803 (.0309)** | **6.558 (.2321)** | **.5843 (.0294)** | 6.517 (.2025) | .5895 (.0255) | **6.648 (.2431)** | **.5728 (.0311)** | **6.623 (.2819)** | **.5758 (.0363)** |
| Raw Data (d=38) | - | - | 7.920(.2545) | .3934(.0392) | **6.514(.3048)** | **.5894(.0382)** | 6.780(.2653) | .5555(.0347) | 6.688(.2465) | .5677(.0319) |

surgery type, and categorical strings (surgeon name, anesthesiologist name and location of the operating room). 23 features were extracted as input from 77,169 surgery cases. The prediction outcome for this dataset is the actual surgery duration. This dataset is referred hereafter as *OR Data*.

In addition, we also evaluate sVAE on the open MIMIC-III (Johnson et al., 2016) for two prediction tasks: a regression task to predict the ICU length of stay (days) and a classification task to predict mortality. Due to space limitation, the supplementary results on the three dataset are reported in the Appendix A4 to A7.

## 5.2 BASELINES

We compared sVAE against state-of-art latent encoding and data transformation methods. For deep generative models, we have included **VAE**, **pi-VAE** that organizes latent space by conditioning on prediction outcome (Zhou & Wei, 2020), **smooth-VAE** (Feng et al., 2020) that represents a broad class of VAEs that express predicted outcome as a latent dimension using the smooth-ELBO objective function in Eq.3, $\beta$-**TCVAE** which is an unsupervised disentangled VAE. Common encoding/transformation techniques such as autoencoder (**AE**), principal component analysis (**PCA**), and independent component analysis (**ICA**) are also implemented. For fair comparison, all neural network-based models are built with same complexity (same number of layers and number of nodes in each hidden layer). As dimension reduction is one of our objective, data were compressed to a lower-dimensional representation ($d = 10$) for all methods.

To evaluate the prediction performance using the encoded/transformed data, several linear and non-linear predictors are built including logistic regression (LR), support vector machine (SVM) with radial basis function kernel, XGBoost, and deep neural network (DNN). The hyper-parameters and implementation are fixed throughout the study and details are described in Appendix A3.

## 5.3 EVALUATION METRICS

We use the following metrics to evaluate the downstream prediction performance, latent disentanglement and input reconstruction. In each experiment, we sample each dataset 5 times randomly and report the mean and standard error of each metric.

**Downstream Prediction:** For classification predictions, we report the area under the Receiver-Operating Characteristic curves (ROC AUC) and the average prediction (equivalent to the area under the Precision Recall curve). The regression performance was evaluated by the root mean square error (rMSE) and $R^2$. As both smooth-VAE and the proposed sVAE explicitly express the predicted outcome as a separate latent dimension, the encoded position on this dimension can be used directly for prediction, by viewing the coordinate on that latent dimension as the predicted values. This is referred as *Direct Prediction* hereafter.

**Latent Disentanglement:** Previous studies (Kim & Mnih, 2018; Chen et al., 2018; Burgess et al., 2018) proposed various metrics for quantifying disentanglement, assuming that we know the factors of variation in the underlying true generative model. This is infeasible in clinical studies. Instead, we use the recently developed Unsupervised Disentanglement Ranking (UDR) (Duan et al., 2019) to assess disentanglement effects. This method relies on the assumption that for a particular dataset and a VAE-based disentangled representation class, disentangled representations are all alike, while every entangled representation is entangled in its own way. Therefore, it can be useful when comparing the latent encoding of models with the same dimensionality, but the metric cannot be used to compare models with different latent dimensionality. When using UDR, higher value implies better disentanglement.

**Input Reconstruction:** Besides prediction performance and disentanglement, it is essential to have a true representation of the original data, such that the disentanglement and interpretation make sense. To evaluate the representation accuracy, we reconstruct the input by decoding the sampled representations from the latent space to the original input space, and the reconstruction error is evaluated by the negative log-liklihood (NLL) of getting the original input from the reconstructed input. A lower NLL implies better accuracy in the reconstruction.

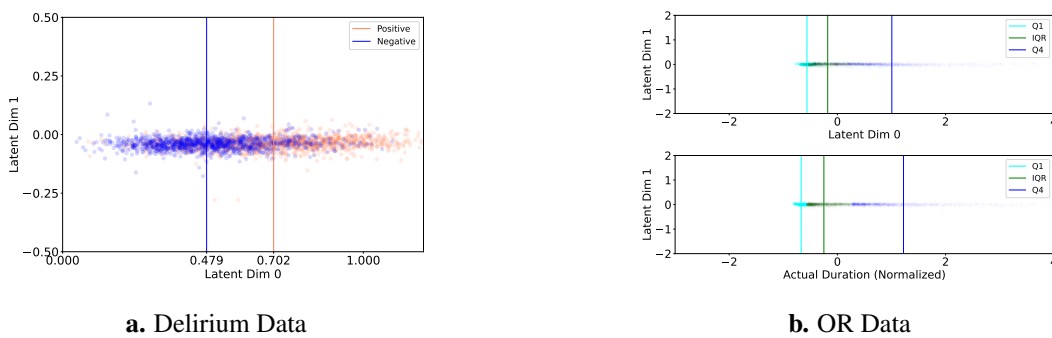

**a.** Delirium Data                    **b.** OR Data

Figure 1: Latent space stratification by prediction tasks in Delirium and OR datasets. In classification prediction (Fig 1.a), cases with positive postoperative delirium are shown in red and cases with negative postoperative delirium are shown in blue. In regression (Fig 1.b), cases are divided into lower quartile (cyan), inter-quartile (green) and upper quartile (blue) of predicted/actual surgery duration. Vertical lines represent the center of mass for each group.

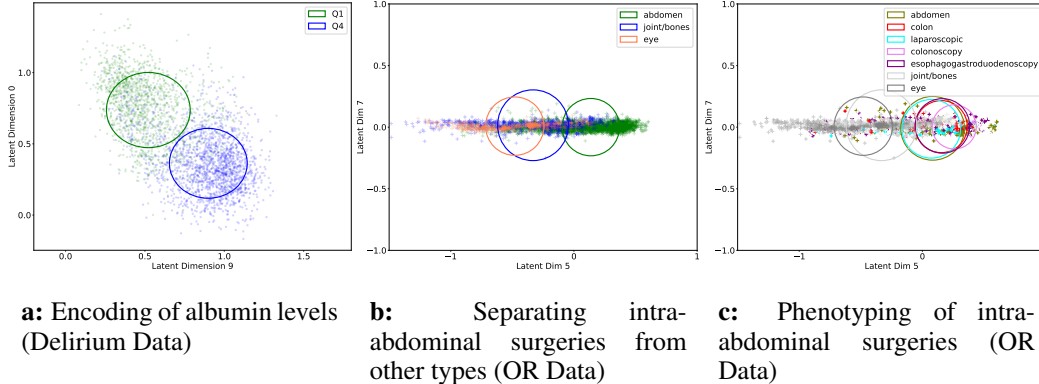

**a:** Encoding of albumin levels (Delirium Data)

**b:** Separating intra-abdominal surgeries from other types (OR Data)

**c:** Phenotyping of intra-abdominal surgeries (OR Data)

Figure 2: Latent space separating patients by phenotypes. Circles represent the cluster of a particular group with radius equal to the average distance between in-class samples and cluster center. a) Information in albumin level is split into two dimensions, as albumin is relevant to prediction. b & c): Latent dimension separating intra-abdominal surgeries from other types while capturing the intrinsic similarity of intra-abdominal surgeries with different free-text descriptions.

## 5.4 QUANTITATIVE COMPARISONS

**sVAE has better prediction performance.** When fixing latent dimensionality to $d = 10$, we observe that sVAE produces consistently better ROC AUC and Average Precision in Delirium data

Table 2: Effects of correlation estimation on prediction, disentanglement and reconstruction of sVAE with varying latent dimensions. Higher values of ROC AUC, Average Precision, and UDR are better, and lower values of Average Pearson Correlation and NLL are better.

| | $d=5$ | $d=10$ | $d=15$ | $d=20$ | $d=25$ | $d=30$ | $d=562$ |
|---|---|---|---|---|---|---|---|
| | Disentanglement: UDR | | | | | | |
| sVAE | **.3018(.0405)** | **.2165(.0227)** | **.1670(.0216)** | **.1490(.0170)** | **.1249(.0108)** | **.1161(.0127)** | - |
| sVAE$^{TC}$ | .2752(.0301) | .1629(.0110) | .1409(.0081) | .1119(.0054) | .0982(.0052) | .0954(.0016) | - |
| | Prediction: ROC AUC | | | | | | |
| sVAE | **.7784(.0085)** | **.7761(.0095)** | **.7784(.0089)** | **.7772(.0067)** | **.7754(.0091)** | **.7769(.0091)** | **.7786(.0094)** |
| sVAE$^{TC}$ | .7577(.0097) | .7581(.0090) | .7576(.0104) | .7585(.0079) | .7553(.0095) | .7574(.0093) | .7573(.0086) |
| | Prediction: Average Precision | | | | | | |
| sVAE | **.7945(.0139)** | **.7943(.0152)** | **.7960(.0171)** | **.7947(.0146)** | **.7949(.0150)** | **.7911(.0158)** | **.7978(.0148)** |
| sVAE$^{TC}$ | .7734(.0156) | .7718(.0146) | .7718(.0171) | .7719(.0159) | .7740(.0161) | .7756(.0146) | .7781(0147) |
| | Reconstruction: Negative Log-likelihood | | | | | | |
| sVAE | **.2056(.3179)** | **.2453(.3149)** | .3811(.4062) | **.1969(.3465)** | **.1214(.1709)** | **.2005(.3481)** | **.3309(.1533)** |
| sVAE$^{TC}$ | .4866(.1132) | .4699(.2611) | **.2783(.2246)** | .3964(.3392) | .4916(.5027) | .3539(.3084) | 1.314(.0912) |

Table 3: Comparison of disentanglement (UDR) of VAE models in three datasets. Higher values in each row imply better disentanglement.

| Dataset | VAE | $\beta$-TCVAE | pi-VAE | smooth-VAE | sVAE |
|---|---|---|---|---|---|
| MIMIC-III | .1088(.0162) | .1164(.0150) | .2156(.0364) | .2501(.0092) | **.3663(.0158)** |
| OR | .1889(.0191) | .2238(.0226) | .3110(.0380) | .2965(.0131) | **.3357(.0111)** |
| Delirium | .1377(.0077) | .1375(.0127) | .1939(.0084) | .1789 (.0059) | **.2165(.0227)** |

(Table 1) and lower rMSE (higher $R^2$) in OR data (Table 2). The prediction performance is even better than predicting on the full-dimensional raw data ($d = 562$ in delirium and $d = 23$ in OR data). Compared to PCA or ICA, encoding/disentangling the raw data ($\beta$-TCVAE) does not benefit the downstream tasks, consistent with previous results by Locatello et al. (2019). Compared to unsupervised AE/VAEs, pi-VAE has slightly better prediction performance, but much lower than models that introduce prediction loss into the loss function (smooth-VAE and sVAE).

**sVAE does not require a predictor.** sVAE distills the predictive information into a single designated dimension and force the other dimensions to be disentangled from this dimension. Consequently, adding a predictor has little improvement over the direct prediction provided by sVAE. This designated prediction dimension is visualized in Fig. 1. For Delirium Data (Fig. 1.a), we force data stratification on latent dimension 0 with respect to the binary outcome of postoperative delirium. As the latent space is generally smooth, the stratification can be viewed analogous to the *predicted* risk of getting delirium. For OR Data (Fig. 1.b), data are stratified with respect to the *predicted* surgery duration. As we minimize the error between prediction and actual duration, the observed distribution on dimension 0 is close to the actual duration. Consistent results are obtained on other datasets (see Appendix A4).

**sVAE has consistently better disentanglement.** To evaluate the disentanglement effects of the proposed correlation estimation metric, we replaced the $TC(\boldsymbol{v})$ by the TC term in $\beta$-TCVAE and the Minibatch-Weighted Sampling proposed by Chen et al. (2018), which is denoted as sVAE$^{TC}$. We compared sVAE$^{TC}$ with sVAE with various dimensionalities. We use Delirium Data as its original input dimensionality is high ($d = 562$), giving us flexibility in adjusting the latent dimensions. We vary the latent dimensionality from 5 to 30. Besides, a full-dimensional latent space ($d = 562$) is also created for reference. At each dimension, we calculate UDR A2A (Duan et al., 2019) for pairwise comparison, and observed that sVAE has consistently better disentanglement than sVAE$^{TC}$ at every dimensionality (see Table 2). As UDR A2A is computationally expensive, the calculation fails for the full-dimension model. To compare with existing baseline VAE models, we have calculated UDR A2A in three datasets and observed that sVAE is consistently better than other VAE models (Table 3). Table 3 also demonstrates the inaccurate estimation of TC when $d = 10$, hence $\beta$-TCVAE does not work well in disentanglement when compared to pi-VAE. Meanwhile, it is noteworthy that the UDR metrics in sVAE and smooth-VAE should not be compared directly with other baseline models, as they have a explicit latent dimension that conveys prediction information, equivalent to a supervised disentanglement for this dimension.

**sVAE has consistent prediction and reconstruction performance with dimension reduction.** Due to the bounded values of $TC(\boldsymbol{v})$, sVAE is able to minimize other loss terms, achieving bet-

ter distillation of predictive information and better input reconstruction. When varying the latent dimensionality, the prediction and reconstruction performance is stable. On the other hand, the wrong estimation of TC term in sVAE$^{TC}$ introduces instability in reconstruction, leading negative log-liklihood (NLL) to 0.4916 when $d = 25$. In extreme cases ($d = 562$), the unbounded TC loss dominates the model training and NLL deteriorates to 1.314.

### 5.5 INTERPRETABILITY THROUGH DISENTANGLEMENT

We now demonstrate through examples that disentanglement enhances the interpretability of the latent representation, which may help clinicians gain trust of the model. Specifically, we show that the model exhibits *phenotyping* effects (Basile & Ritchie, 2018), capturing the intrinsic characteristics of patients from the noisy input data, showing patterns in agreement with the existing clinical evidence. Here we give two examples from Delirium Data and OR Data, respectively. Examples of MIMIC-III are in Appendix A5.

In Delirium Data, the input variable *albumin level* is noteworthy, as clinical evidence showed that it is associated with postoperative delirium (Zhang et al., 2018). If sVAE captures the intrinsic association between albumin and delirium, then dimension 0 (representing the delirium prediction) should contribute to the distribution of albumin levels in the latent space. As shown in Fig. 2a, the distribution of albumin values are controlled by both dimension 0 and another dimension, in agreement with clinical expectation.

In OR Data, the most complex variable is planned surgery description consisting of free texts with varying lengths, formats and unsystematic expressions. It usually requires human efforts to read and interpret the notes before billing and staffing. Among the surgical description texts, the most frequent surgery types are various types of intra-abdominal surgeries ( *abdomen, colon, esophagogastroduodenoscopy, laparoscopic, and colonoscopy*), occurring in from 16.75% to 4.16% of test cases. Besides, eye-related surgery constitutes 16.75% and bone/joint related keywords (*knee, leg, spine, anthroplasty*) constitutes 9.54% in total. The various types of intra-abdominal surgeries are all performed on soft tissues in abdomen regions. They generally have different indications, recovery profiles, and complications from orthopedic or ophthalmic surgeries. They usually also last longer than orthopedic or ophthalmic surgeries. In addition, the inclusion of surgeon name as an input variable provides another reason for there to be a latent class dividing surgeries based on the part of the body, as most surgeons specialize in operating on a particular organ. In the latent space, we observe that latent dimension 5 represents the characteristics of intra-abdomenal surgeries, and separates them from eye-related or joint/bone surgeries (Fig. 2b). We further split abdomen surgeries with respect to specific keywords, with each circle representing the cluster of a keyword and radius representing the average distance between cases and the cluster center. As shown in Fig 2.c keywords representing similar surgery types/regions related to abdomen are clustered to the right, with little overlap to eye or joint/bone related surgeries. This implies that latent dimension 5 learns the intrinsic information underlying the different textual expressions, when different keywords are used to describe intra-abdomen regions.

## 6 CONCLUSION

In this paper, we present sVAE, a prediction-based, interpretable deep latent model. sVAE provides a holistic framework that integrates regularization-based disentanglement and prediction-guided encoding. We apply sVAE to two real-world perioperative datasets and evaluate its efficacy and performance in the context of surgical applications. Our results demonstrate sVAE's latent prediction dimension can be used directly for prediction with superior performance over predictors based on state-of-the-art VAE models. Furthermore, the disentanglement of latent spaces demonstrate the phenotyping effects by capturing the intrinsic characteristics of clinical data, and create clinically meaningful organizations from the noisy inputs. While sVAE was motivated by clinical applications, the proposed approach makes general advancement toward future adoption of VAE models in fields where predictive performance and interpretability are essential in the presence of high-dimensional and noisy data.

## 7 ETHICS STATEMENT AND REPRODUCIBILITY

There are three datasets involved in this application study. Delirium Data and OR Data were acquired with appropriate IRB approval of the study with waivers of consent. These clinical datasets were not deidentified or approved for publication. For reproducibility, the same experiments were repeated on an open clinical dataset MIMIC-III (Johnson et al., 2016) and the supplemtary results were attached in Appendix A4-7. Detailed description of model hyper-parameters and Python code repository can be found in Appendix A3.

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

# A    APPENDIX

## A.1    ESTIMATION OF CORRELATION

Generate a latent sample from $\boldsymbol{v}^{(i)} \sim q_{\phi'}(\boldsymbol{v}|x^{(i)})$, where $i = 1, ..., B$ are the indices of samples of current batch data. Standardize and concatenate into a $B \times d$ matrix $\boldsymbol{A} = [\boldsymbol{v}_1, \boldsymbol{v}_2, ..., \boldsymbol{v}_B]^T$. As each latent dimension may have different mean/variance, each dimension should be standardized, yielding $\boldsymbol{A}$. Perform singular value decomposition, we get $\boldsymbol{A} = \boldsymbol{U}\boldsymbol{\Sigma}\boldsymbol{V}^T = \sum_{i=1}^{d} \sigma_i \boldsymbol{u}_i \boldsymbol{v}_i^T$, where $\sigma_1 \geq \sigma_2 \geq ... \geq \sigma_d \geq 0$. Then $TC(\boldsymbol{v})$ has 3 key properties:

1.  $TC(\boldsymbol{v}) = \frac{\sigma_1^2 - 1}{d - 1}$ is bounded by $[0, 1]$;

2.  $TC(\boldsymbol{v})$ is minimized iff the all latent dimensions are uncorrelated (orthogonal);

3.  $TC(\boldsymbol{v}) = TC(v^{(1)}, v^{(2)}, ..., v^{(B)})$ is the lower bound of the $TC$ of the whole latent space, calculated by $TC(v^{(1)}, v^{(2)}, ..., v^{(N)})$. Meanwhile, the minimization across all batches minimizes the upper bound of the whole latent space.

### A.1.1    PROPERTY 1

To see 1, note that $\boldsymbol{A}$ is standardized therefore by definition we get $||\boldsymbol{A}||_F^2 = \sum_{i=1}^{d} \sum_{j=1}^{n} A_{i,j}^2 = d$. Besides, it is easy to show that

$$||\boldsymbol{A}||_F^2 = \sum_{i=1}^{d} \sigma_i^2 \tag{6}$$

which implies that for the largest singular value we have $1 \leq \sigma_i \leq \sqrt{d}$.

### A.1.2    PROPERTY 2

To see 2: • when $TC(\boldsymbol{v})$ is minimized, $\sigma_1 = 1$. As $\sigma_1$ is the largest singular values and by Eq. 9, we get $\sigma_1 = \sigma_2 = ... = \sigma_d = 1$. Therefore, $\boldsymbol{A} = \boldsymbol{U}\boldsymbol{\Sigma}\boldsymbol{V}^T = \boldsymbol{U}\boldsymbol{V}^T$. Since $\boldsymbol{A}$ is now a product of orthogonal matrices hence itself is also orthogonal. • In the reverse direction, if all latent dimensions are uncorrelated, we get $\boldsymbol{A}\boldsymbol{A}^T = 1$, so the singular values of $\boldsymbol{A}$ are all 1 and $TC(\boldsymbol{v}) = 0$.

For the property 2, intuitively $TC(\boldsymbol{v})$ measures the maximum variance that can be explained by a principal component. When the largest principle component can only explain minimum variance, we know that the latent space is decorrelated (disentangled).

### A.1.3    PROPERTY 3

To prove 3, we first show that $TC(\boldsymbol{v})$ is the lower bound. Note that the singular values of $\boldsymbol{A}$ are square roots of the eigenvalues of $\boldsymbol{A}^T \boldsymbol{A}$. When estimating the whole latent space ($B = N$), this is equivalent to

$$\begin{pmatrix} \boldsymbol{A}^T & \boldsymbol{X}^T \end{pmatrix} \begin{pmatrix} \boldsymbol{A} \\ \boldsymbol{X} \end{pmatrix} = \boldsymbol{A}^T \boldsymbol{A} + \boldsymbol{X}^T \boldsymbol{X} \tag{7}$$

where X are the latent variables of data not in the current batch ($\boldsymbol{X}$ contains $N - B$ rows and $d$ columns). So the question is equivalent to estimating the eigenvalues of the sum of symmetric (in this case also positive semidefinite) matrices using only $\boldsymbol{A}^T \boldsymbol{A}$. From Courant-Fischer min-max theorem, we know that the eigenvalues of $\boldsymbol{A}^T \boldsymbol{A}$ are:

$$\lambda_k(\boldsymbol{A}^T \boldsymbol{A}) = \min\{\max\{R_{\boldsymbol{A}^T \boldsymbol{A}}(\boldsymbol{x})|\boldsymbol{x} \in \boldsymbol{U} \text{ and } \boldsymbol{x} \neq 0\}| \dim(\boldsymbol{U}) = k\}$$

where $R_{\boldsymbol{A}^T \boldsymbol{A}}(\boldsymbol{x}) = \frac{(\boldsymbol{A}^T \boldsymbol{A}\boldsymbol{x}, \boldsymbol{x})}{(\boldsymbol{x}, \boldsymbol{x})}$.

Since $R_{\boldsymbol{A}^T \boldsymbol{A} + \boldsymbol{X}^T \boldsymbol{X}}(\boldsymbol{x}) \geq \max\{R_{\boldsymbol{A}^T \boldsymbol{A}}(\boldsymbol{x}), R_{\boldsymbol{X}^T \boldsymbol{X}}(\boldsymbol{x})\}$, we know that $\lambda_k(\boldsymbol{A}^T \boldsymbol{A}) \leq \lambda_k(\boldsymbol{A}^T \boldsymbol{A} + \boldsymbol{X}^T \boldsymbol{X})$, hence $TC(\boldsymbol{v})$ forms the lower bound of the true $TC(\boldsymbol{v})$ when estimated using whole data.

Next, we show that after each epoch, we have minimized the upper bound of the actual eigenvalues of the true data. Denote the first batch of data by $\boldsymbol{A}_1$, and then the whole data can be expressed as

$$\boldsymbol{D} = \begin{pmatrix} \boldsymbol{A}_1 \\ \boldsymbol{X}_1 \end{pmatrix} \tag{8}$$

, we know that $\boldsymbol{D}^T\boldsymbol{D} = \boldsymbol{A}_1^T\boldsymbol{A}_1 + \boldsymbol{X}_1^T\boldsymbol{X}_1$. Then

$$TC(\boldsymbol{D}) = TC(\boldsymbol{A}_1^T\boldsymbol{A}_1 + \boldsymbol{X}_1\boldsymbol{X}_1^T)$$

hence

$$
\begin{aligned}
TC(\boldsymbol{D}) = \frac{\sqrt{||\boldsymbol{D}^T\boldsymbol{D}||_2} - 1}{d-1} &\leq \frac{\sqrt{||\boldsymbol{A}_1^T\boldsymbol{A}_1||_2} - 1}{d-1} + \frac{\sqrt{||\boldsymbol{X}_1\boldsymbol{X}_1^T||_2} - 1}{d-1} \\
&\leq \frac{\sqrt{||\boldsymbol{A}_1^T\boldsymbol{A}_1||_2} - 1}{d-1} + \frac{\sqrt{||\boldsymbol{A}_2\boldsymbol{A}_2^T||_2} - 1}{d-1} + \frac{\sqrt{||\boldsymbol{X}_2\boldsymbol{X}_2^T||_2} - 1}{d-1} \\
&... \\
&= TC_1(\boldsymbol{v}) + TC_2(\boldsymbol{v}) + ... + TC_T(\boldsymbol{v}))
\end{aligned}
$$

After a complete epoch looping through all training data, we have actually minimized the upper-bound the the true estimation. When the training converges, we would know that $\max\{TC_i(\boldsymbol{v})\}_1^T \leq TC(\boldsymbol{D}) \leq \operatorname{sum}\{TC_i(\boldsymbol{v})\}_1^T$.

## A.2 GRAPHICAL DIAGRAM OF sVAE

As raw EHR data consists of non-numerical or non-static variables, a standardized data-processing was conducted for each dataset to get the static, numerical representation of $X$ as illustrated in Fig. A1.

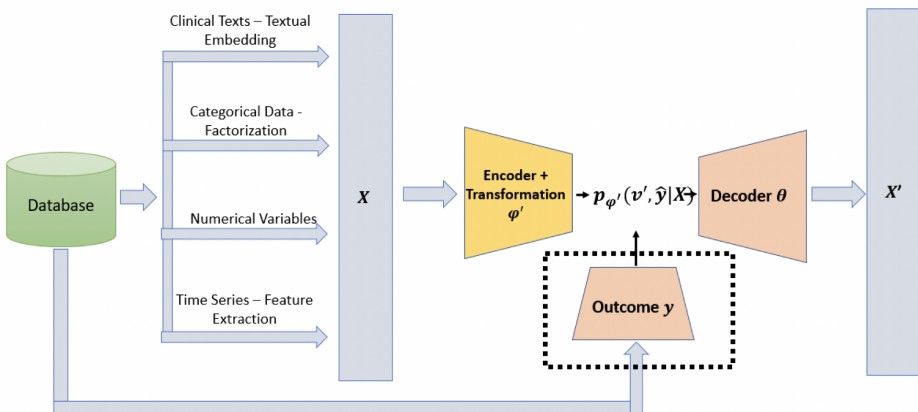

Figure A1: Graphical diagram of sVAE. Heterogeneous mixed type data are first converted to numerical embedding, and then fed into the encoder. Prediction task is only visible by sVAE during the training stage to regulate the latent space, and is not needed when encoding the testing data
.

### A.3 HYPER-PARAMETERS AND IMPLEMENTATION DETAILS

All deep latent models, including AE, VAE, $\beta$-TCVAE, smooth-VAE, and sVAE are implemented using Python Pytorch (Paszke et al., 2019), with the same configuration for all datasets and models. Essential parameters include: activation function = exponential linear unit (ELU), optimizer = Adam (learning rate= $1e^{-6}$), batchsize = 256, number of hidden layers = 4 (same for both encoder and decoder), and number of hidden units (10*d, h, 4*h, h) for decoder and (h, 8*h, 4*h, h) for encoder, where h is a hyper-parameter. Currently h = 2000. The DNN predictor is also using Pytorch with the same set of parameters as every encoder.

Other predictors (LR, XGBoost and SVM) and transformation methods (PCA and ICA) are implemented by Sklearn package Pedregosa et al. (2011). For LR, we used l1 penalty and linear solve with regularzier C=0.2. For Xgboost, the number of estimators for XGBoost is 10 (same as the latent dimension) with loss as "ls" in regression and "binary:logistic" in classification. LogisticRegression(penalty='l1',solver='liblinear',C=0.2). For SVM, the ROC AUC and average precision are calculated by the decision function.

Implementation and an example running report using Jupyter Notebook are here https://anonymous.4open.science/r/sVAE-E2DB/.

## A.4 SUPPLEMENTARY EXPERIMENT RESULTS OF ALL DATASETS

**MIMIC-III**: Each of the above-mentioned datasets only has a single prediction outcome associated to the inputs, and are not publicly available. To extend our experiment to the scenario of multi-task training on a public dataset, we have conducted experiment on the clinical dataset MIMIC-III Johnson et al. (2016). Out of the 53,423 distinct hospital admissions for adult patients in MIMIC-III, we identified 57,786 valid distinct ICU stays in total, where multiple stays might have occurred during the same admission. Input variables include demographics, history of diseases from ICD codes, and ICD procedure descriptions, converted to 40 numerical features after pre-processing, with missing rate approximately 11.79%. Two prediction outcomes are extracted from MIMIC-III for all included cases: a regression task to predict the ICU length of stay (days) and a classification task to predict mortality. This dataset is referred hereafter as "MIMIC-III".

Table A1: Classification performance for postoperative delirium prediction.

| Transformation Method (d=10) | Direct Prediction | | LR | | XGBoost | | SVM | | DNN | |
|---|---|---|---|---|---|---|---|---|---|---|
| | ROC AUC | Average Precision | ROC AUC | Average Precision | ROC AUC | Average Precision | ROC AUC | Average Precision | ROC AUC | Average Precision |
| PCA | - | - | .7168(.0091) | .7394 (.0151) | .7062 (.0128) | .7284 (.0109) | .6299 (.0713) | .6522 (.0765) | .7221 (.0084) | .7462 (.0153) |
| ICA | - | - | .7466(.0089) | .7695 (.0122) | .6724 (.0066) | .7013 (.0123) | .6878 (.0016) | .7048 (.0090) | .7455 (.0086) | .7665 (.0127) |
| AE | - | - | .6431(.0075) | .6571 (.0149) | .6406 (.0088) | .6526 (.0104) | .6452 (.0090) | .6595 (.0129) | .6367 (.0105) | .6518 (.0136) |
| VAE | - | - | .6473(.0066) | .6667 (.0090) | .6503 (.0099) | .6677 (.0107) | .6486 (.0062) | .6683 (.0145) | .6348 (.0089) | .6540 (.0117) |
| $\beta$-TCVAE | - | - | .6563(.0104) | .6722 (.0071) | .6553 (.0109) | .6723 (.0061) | .6565 (.0092) | .6712 (.0082) | .6383 (.0148) | .6562(.0126) |
| pi-VAE | - | - | .6900(.0139) | .7135 (.0182) | .6669(.0108) | .6814 (.0163) | .6871 (.0173) | .7097 (.0215) | .6684 (.0119) | .6883(.0236) |
| smooth-VAE | .7588 (.0103) | .7762 (.0147) | .7599 (.0100) | .7781 (.0149) | .7584 (.0100) | .7725 (.0140) | .7598 (.0096) | .7773 (.0133) | .7598 (.0097) | .7782 (.0146) |
| sVAE | **.7761 (.0095)** | **.7943 (.0152)** | **.7725 (.0101)** | **.7900(.0174)** | **.7738 (.0091)** | **.7873(.0115)** | **.7691 (.0097)** | **.7862(.0163)** | **.7746 (.0091)** | **.7926(.0161)** |
| Raw Data (d=562) | - | - | .7576(.0091) | .7798 (.0153) | .7374 (.0100) | .7640 (.0127) | .7459 (.0071) | .7691 (.0149) | .7422 (.0111) | .7629 (.0146) |

Table A2: Regression performance for predicting length of stay in operating room.

| Transformation Method (d=10) | Direct Prediction | | LR | | XGBoost | | SVM | | DNN | |
|---|---|---|---|---|---|---|---|---|---|---|
| | rMSE | $R^2$ | rMSE | $R^2$ | rMSE | $R^2$ | rMSE | $R^2$ | rMSE | $R^2$ |
| PCA | - | - | 94.33(2.915) | .4806(.0320) | 85.83(3.014) | .5699(.0300) | 78.17(3.593) | .6429(.0325) | 82.50(2.874) | .6027(.0275) |
| ICA | - | - | 111.4 (9.296) | .2717(.1190) | 96.74 (7.782) | .4508 *.0838) | 129.0 (1.702) | .0295(.0258) | 120.8 (7.114) | .1462 (.0982) |
| AE | - | - | 94.97(1.845) | .4739(.0205) | 88.65(2.098) | .5415(.0216) | 86.93(3.215) | .5588(.0327) | 84.49(1.966) | .5836(.0194) |
| VAE | - | - | 100.4(2.821) | .411(.0331) | 100.9 (3.510) | .4056(.0403) | 106.7(2.985) | .3357(.0373) | 105.99(3.230) | .3443(.0402) |
| $\beta$-TCVAE | - | - | 103.4 (5.523) | .3749(.0660) | 95.72 (8.412) | .4617(.0930) | 109.3 (5.521) | .3010(.0698) | 107.9 (6.593) | .3189(.0844) |
| pi-VAE | - | - | 98.80(6.359) | .4284(.0759) | 92.74 (5.875) | .4965(.0648) | 79.45(7.640) | .6285(.0722) | 85.22 (5.331) | .5748 (.0536) |
| smooth-VAE | 68.34 (2.837) | .7272(.0226) | 68.17 (2.793) | .7286(.0222) | **68.09 (3.065)** | **.7292(.0244)** | 68.28 (2.749) | .7277(.0219) | 69.04 (2.454) | .7217(.0197) |
| sVAE | **67.33(3.159)** | **.7351(.0249)** | **67.11(3.199)** | **.7368(.0251)** | 68.98(3.090) | .7220(.0249) | **66.95(3.172)** | **.7380(.0248)** | **68.26(4.067)** | **.7273(.0328)** |
| Raw Data (d=23) | - | - | 83.60 (3.049) | .5919 (.0297) | 68.36 (2.645) | .7271 (.0210) | 71.32 (3.519) | .7027 (.0290) | 69.51 (2.353) | .7179 (.0191) |

Table A3: Regression performance for MIMIC-III ICU Length-of-Stay prediction. Mean and standard error are reported after 5 random shuffles.

| Transformation Method (d=10) | Direct Prediction | | LR | | XGBoost | | SVM | | DNN | |
|---|---|---|---|---|---|---|---|---|---|---|
| | rMSE | $R^2$ | rMSE | $R^2$ | rMSE | $R^2$ | rMSE | $R^2$ | rMSE | $R^2$ |
| sVAE$^{TC}$ | 6.719 (.2649) | .5635 (.0344) | 6.686 (.2560) | .5679 (.0330) | 6.615 (.2356) | .5770 (.0299) | 6.780 (.2550) | .5556 (.0333) | 6.748 (.2103) | .5600 (.0274) |
| sVAE | **6.604 (.2260)** | **.5786 (.0288)** | **6.551 (.2336)** | **.5852 (.0296)** | 6.534 (.1856) | .5876 ( .0233) | **6.647 (.2426)** | **.5729 (.0311)** | **6.640 (.2203)** | **.5739 (.0283)** |
| Raw Data (d=38) | - | - | 7.920(.2545) | .3934(.0392) | **6.514(.3048)** | **.5894(.0382)** | 6.780(.2653) | .5555(.0347) | 6.688(.2465) | .5677(.0319) |

Table A4: Classification performance for MIMIC-III mortality prediction. Mean and standard error are reported after 5 random shuffles.

| Transformation Method (d=10) | Direct Prediction | | LR | | XGBoost | | SVM | | DNN | |
|---|---|---|---|---|---|---|---|---|---|---|
| | ROC AUC | Average Precision | ROC AUC | Average Precision | ROC AUC | Average Precision | ROC AUC | Average Precision | ROC AUC | Average Precision |
| sVAE$^{TC}$ | .8337(.0023) | .7299(.0055) | .8336(.0023) | .7296(.0058) | .8333(.0026) | .7180(.0055) | .8337(.0025) | .7297(.0058) | .8335(.0024) | .7296(.0055) |
| sVAE | **.8365(.0021)** | **.7333(.0046)** | **.8372(.0020)** | **.7345(.0054)** | **.8359(.0022)** | **.7208(.0039)** | **.8370(.0017)** | **.7343(.0043)** | **.8372(.0020)** | **.7345(.0054)** |
| Raw Data (d=38) | - | - | .7959(.0038) | .6759(.0059) | .8248 (.0024) | .7214 (.0041) | .8022 (.0000) | .6812 (.0000) | .8358 (.0029) | .7323 (.0036) |

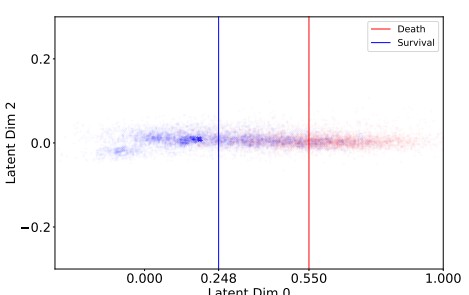 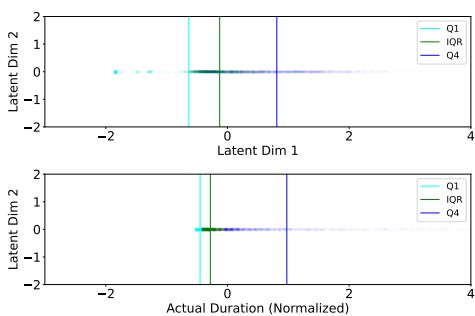

**b:** Latent space organization by predicted risk of mortality (Dataset: MIMIC-III)

**d:** Latent space organization by predicted ICU length of stay (Dataset: MIMIC-III)

Figure A2: Latent space stratification by prediction task in multi-task training. Positive cases in mortality prediction are shown in red and negative cases in blue. For regression prediction of ICU LoS, cases are divided into lower quartile (cyan), inter-quartile (green) and upper quartile (blue). Vertical lines represent the center of mass for each group.

## A.5 LATENT SPACE INTERPRETATION: MIMIC-III

In MIMIC-III, we show that the model is able to learn the intrinsic information of ICD procedure codes with the promising application to billing process, as most procedures are used to treat conditions that affect a particular organ (e.g., mechanical ventilation is used to treat acute respiratory failure). It makes sense for an ICD procedure code and an ICD diagnosis code to point to dysfunction of a particular organ. In the 11,716 hold-out test cases, 'ventilation', 'arteriography', 'artery' and 'lung' are frequent clinical keywords, occurring in 27.71%, 14.81%, 14.60% and 4.33% cases respectively. In latent space, dimension 5 and dimension 10 jointly represents the relevant information, and well-separate these two groups. Meanwhile, the similarity between 'arteriography' and 'artery' and between 'artery' and 'lung' are captured, so that ICD procedures with similar meaning but different keywords are captured and clustered in all latent dimensions.

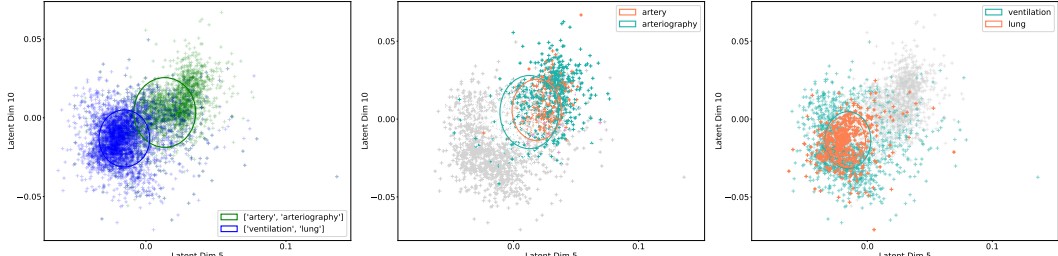

**a:** Latent dimension separating lung-disease ICU patients from artery-related ICU patients

**b:** Phenotyping of patients with artery-related ICD procedure codes

**c:** Phenotyping of patients with lung-related ICD procedure codes

**a & b & c:** Latent dimension separating ICU patients with lung-related problems from artery-related problems, meanwhile it captures the intrinsic characteristics within each group, although described with different ICD codes in input data (Dataset: MIMIC-III)

Figure A3: Latent space separating patients by phenotypes. Circles represent the cluster of a particular group with radius equal to the average distance between in-class samples and cluster center.

A.6 ESTIMATION OF TC IN TRAINING

To illustrate the unboundedness of KL-based method in the minimization of ELBO function, we have recorded the average loss of TC term after running for 100 epochs for various latent dimensionality.

As directly estimating the TC term is intractable, we used the estimating by Minibatch Sampling proposed by Chen et al. (2018). Two sampling strategies, Minibatch Stratified Sampling (MSS) performed indifferently from Minibatch Weighted Sampling (MWS) as shown in C2.2 in Chen et al. (2018), hence we randomly pick MWS to record the estimated loss on training data. For reference, we also calculated the average Pearson's correlation coefficient between all pairs of dimensions. As we can see, when dimensionality does larger, the estimation of TC is not lower bounded; due to the curse of dimensionality, this inevitably gets worse.

Table A5: Unboundness of TC estimation (log) in training

| Dimensionality | TC (MSS) | Avg. Pearson Correlation |
|---|---|---|
| 5 | -15.68 (.0513) | .0349(.0169) |
| 10 | -35.24 (.1103) | .0253(.0246) |
| 15 | -54.69(.1761) | .0406(.0211) |
| 20 | -74.19(.2299) | .0367(.0309) |
| 25 | -93.35(.6048) | .0944(.1036) |
| 30 | -112.7 (.4669) | .0729 (.0360) |
| 562 | -2175(7.208) | .0643(.0182) |

## A.7 MORE COMPARISON WITH BASELINES

We have extended Table 3 to all baseline variants of VAE models, and tabulted their classification performance (Table A4 and A5), disentanglement and reconstruction error (Table A6) below. As we can see, our proposed sVAE continues to outperform baseline models in prediction and disentanglement, irrespective of the latent dimensionality. In reconstruction, it is noteworthy that $\beta$-TCVAE has the best reconstruction performance than other VAE models.

Table A6: ROC AUC of baseline methods with respect to dimension.

| | d=5 | d=10 | d=15 | d=20 | d=25 | d=30 | d=562 |
|---|---|---|---|---|---|---|---|
| | pi-VAE | | | | | | |
| LR | .6748(.0075) | .6920(.0149) | .7022(.0205) | .7158(.0094) | .7243(.0103) | .7266(.0094) | .7621(.0079) |
| GBT | .6623(.0120) | .6649(.0112) | .6739(.0282) | .6796(.0101) | .6912(.0153) | .6933(.0177) | .7244(.0080) |
| SVM | .6619(.0259) | .6676(.0287) | .7008(.0192) | .7143(.0084) | .7173(.0123) | .7225(.0109) | .7377(.0334) |
| DNN | .6632(.0087) | .6688(.0109) | .6739(.0199) | .6868(.0119) | .6948(.0144) | .6953(.0124) | .7583(.0065) |
| | $\beta$-TCVAE | | | | | | |
| LR | .6364(.0149) | .6561(.0117) | .6659(.0191) | .6743(.0161) | .6787(.0142) | .6812(.0102) | .7141(.0054) |
| GBT | .6415(.0161) | .6569(.0117) | .6651(.0136) | .6703(.0105) | .6686(.0098) | .6711(.0048) | .6671(.0100) |
| SVM | .6373(.0144) | .6563(.0103) | .6680(.0183) | .6785(.0150) | .6836(.0151) | .6874(.0089) | .7404(.0072) |
| DNN | .6252(.0106) | .6387(.0167) | .6539(.0178) | .6621(.0146) | .6671(.0182) | .6774(.0083) | .7362(.0053) |
| | smooth-VAE | | | | | | |
| Direction Prediction | .7558(.0133) | .7608(.0106) | .7562(.0082) | .7587(.0095) | .7544(.0091) | .7568(.0091) | .7579(.0079) |
| LR | .7579(.0122) | .7616(.0105) | .7576(.0087) | .7609(.0090) | .7572(.0090) | .7596(.0089) | .7588(.0081) |
| GBT | .7557(.0125) | .7603(.0103) | .7547(.0087) | .7594(.0096) | .7547(.0096) | .7575(.0089) | .7570(.0072) |
| SVM | .7578(.0121) | .7615(.0102) | .7567(.0087) | .7605(.0086) | .7557(.0104) | .7582(.0095) | .7549(.0092) |
| DNN | .7588(.0118) | .7614(.0103) | .7586(.0083) | .7612(.0086) | .7580(.0087) | .7603(.0082) | .7589(.0091) |

Table A7: Average Precision of baseline methods with respect to dimension.

| | d=5 | d=10 | d=15 | d=20 | d=25 | d=30 | d=562 |
|---|---|---|---|---|---|---|---|
| | pi-VAE | | | | | | |
| LR | .6950(.0165) | .7152(.0201) | .7193(.0250) | .7378(.0067) | .7511(.0145) | .7519(.0139) | .7845(.0147) |
| GBT | .6781(.0044) | .6777(.0163) | .6890(.0195) | .6984(.0068) | .7090(.0151) | .7144(.0184) | .7468(.0129) |
| SVM | .6785(.0193) | .6884(.0395) | .7225(.0247) | .7332(.0075) | .7410(.0174) | .7486(.0144) | .7572(.0386) |
| DNN | .6787(.0142) | .6901(.0267) | .6928(.0241) | .7029(.0042) | .7214(.0158) | .7165(.0216) | .7802(.0124) |
| | $\beta$-TCVAE | | | | | | |
| LR | .6571(.0236) | .6738(.0070) | .6858(.0121) | .6999(.0282) | .6997(.0144) | .7010(.0134) | .7397(.0123) |
| GBT | .6619(.0206) | .6735(.0063) | .6833(.0074) | .6898(.0171) | .6905(.0113) | .6942(.0108) | .6901(.0067) |
| SVM | .6560(.0229) | .6722(.0090) | .6880(.0094) | .7025(.0262) | .7009(.0128) | .7054(.0145) | .7606(.0174) |
| DNN | .6429(.0176) | .6589(.0120) | .6731(.0126) | .6888(.0225) | .6868(.0170) | .6985(.0173) | .7576(.0173) |
| | smooth-VAE | | | | | | |
| Direction Prediction | .7746(.0164) | .7787(.0155) | .7739(.0144) | .7761(.0143) | .7723(.0145) | .7762(.0150) | .7763(.0141) |
| LR | .7760(.0153) | .7803(.0159) | .7756(.0156) | .7791(.0136) | .7758(.0158) | .7796(.0150) | .7775(.0133) |
| GBT | .7689(.0145) | .7747(.0153) | .7698(.0154) | .7750(.0139) | .7709(.0137) | .7750(.0103) | .7753(.0122) |
| SVM | .7754(.0153) | .7793(.0142) | .7748(.0165) | .7784(.0133) | .7739(.0179) | .7766(.0161) | .7740(.0149) |
| DNN | .7772(.0153) | .7803(.0155) | .7773(.0143) | .7796(.0131) | .7776(.0163) | .7800(.0146) | .7769(.0165) |

Table A8: Effects of correlation estimation on prediction, disentanglement and reconstruction of sVAE with varying latent dimensions. Higher values of ROC AUC, Average Precision, and UDR are better, and lower values of Average Pearson Correlation and NLL are better.

| | d=5 | d=10 | d=15 | d=20 | d=25 | d=30 | |
|---|---|---|---|---|---|---|---|
| | Disentanglement: UDR | | | | | | |
| pi-VAE | .2937(.0211) | .1939(.0084) | .1372(.0204) | .1161(.0088) | .1073(.0055) | .0945(.0070) | - |
| $\beta$-TCVAE | .1587(.0189) | .1375(.0127) | .1138(.0055) | .0879(.0068) | .0814(.0036) | .0811(.0064) | - |
| smooth-VAE | .2726(.0222) | .1789(.0059) | .1585(.0093) | .1170(.0050) | .1171(.0036) | .0979(.0026) | - |
| | Reconstruction: Negative Log-likelihood | | | | | | |
| pi-VAE | .5896(.8472) | .4037(.4903) | .2962(.4417) | .5047(.8062) | .5936(.9701) | .4261(.6888) | .4169(.4977) |
| $\beta$-TCVAE | .1356(.0242) | .1684(.1364) | .2399(.2902) | .1898(.2258) | .0802(.0553) | .0991(.1142) | 1.353(2.242) |
| smooth-VAE | .3533(.2376) | .6331(.1380) | .4754(.3935) | .3151(.3333) | .2306(.1938) | .2237(.3054) | .5271(.6684) |

## A.8 GRID SEARCH OF HYPER PARAMETERS

Table A9: Classification performance when varying hyper parameters (MIMIC-III mortality prediction). $w$: weight of prediction loss; $\beta$: weight of disentanglement.

| Hyper Parameters $(w, \beta)$ | $w = 1$ | | $w = 4$ | | $w = 7$ | | $w = 10$ | | $w = 13$ | | $w = 16$ | | $w = 19$ | | $w = 22$ | |
|---|---|---|---|---|---|---|---|---|---|---|---|---|---|---|---|---|
| | ROC AUC | Average Precision | ROC AUC | Average Precision | ROC AUC | Average Precision | ROC AUC | Average Precision | ROC AUC | Average Precision | ROC AUC | Average Precision | ROC AUC | Average Precision | ROC AUC | Average Precision |
| $\beta = 1$ | .8402 | .7210 | .8381 | .7197 | .8399 | .7218 | .8386 | .7199 | .8406 | .7215 | **.8412** | **.7274** | .8378 | .7194 | .8391 | .7253 |
| $\beta = 4$ | .8387 | .7218 | .8391 | .7217 | .8390 | .7194 | .8405 | .7250 | .8406 | .7228 | .8388 | .7182 | .8385 | .7178 | .8399 | .7218 |
| $\beta = 7$ | .8385 | .7179 | .8371 | .7218 | .8390 | .7199 | .8396 | .7237 | .8391 | .7266 | .8410 | .7219 | .8373 | .7209 | .8389 | .7193 |
| $\beta = 10$ | .8390 | .7214 | .8377 | .7173 | .8403 | .7216 | .8382 | .7210 | .8391 | .7266 | .8410 | .7219 | .8373 | .7209 | .8389 | .7193 |

