# OpenReview forum: "Surgical Prediction with Interpretable Latent Representation"
_ICLR.cc/2022/Conference — ICLR 2022 Submitted_

### Official Review · Reviewer_56Ru · 2021-10-20

**Correctness:** 3
**Technical Novelty And Significance:** 2
**Empirical Novelty And Significance:** 1
**Recommendation:** 3
**Confidence:** 4

**Main Review:**

In addition to comments made in the Summary of the Paper section, some experimental issues exist. How the (severe) missing rate in the delerium data was handled is not expressed. It is good to include MIMIC-III, and the baselines are well chosen (although how they were optimized is also not expressed). Instead of choosing a single standard for all baselines, it would have been preferable to understand their strengths and weaknesses at different hyper-parameters (e.g., does one’s performance grow more or less quickly with more (or fewer) reduced features $d$? I.e., Table 3 but with all baselines). Gaussians are used w/o consideration, but this is as usual so it is not taken as a negative. Although sVAE clearly outperforms many baselines, the degree to which it’s close to smooth-VAE, especially considering (supposed?) variances would make a significance test, and some cross-validation, advisable.

The interpretability angle is interesting and appreciated, especially within the medical context. Not enough has been shown to warrant the mention of ‘trust’ in the conclusion, though.

There are a few textual issues, including grammar (e.g., “… we want our sVAE to … minimizes …”) but in general the text is well structured and constructed.

**Summary Of The Paper:**

This paper proposes a representation learning framework for high-dimensional, complex clinical data. It is not clear to what extent this is specific to *surgery*, however. The related works section is appropriately structured, deep, and recent. Some of the text (e.g., around background) is needlessly simplistic.

The primary methodological contribution orbits around the function $g()$— the  disentangling function applied to the latent representation of an encoder. Naturally, there are many papers that disentangle latent representations (e.g., Yan et al 2016 (Attribute2image), Higgins et al 2017 (beta-VAE), Mathieu et al 2019), including various attempts at sequential encoding; some of this work is cited in Sec 2, and the efficient computation of the total correlation is presented as the key step to add and then overcome. The result is Eq 6 and SVD is subsequently used in the estimation/replacement of TC. While the rationale provided is _fine_, some claims made regarding the intuition for this step do not seem to be matched by the rationale/citations given, and it would have been advisable to dig into this substation theoretically, a bit more.

**Summary Of The Review:**

Generally, there are no major concerns with regards to the empirical setup (nothing that was done appears incorrect or suggests incorrect results), except that more should have been done to validate the results. The technical contribution over the vast amount of other work on disentangling latent representations is clear but perhaps slight and therefore the applied angle seems to be emphasized (indeed, ‘surgery’ is even in the name of the model) but, apart from using some relevant datasets and showing some qualitative results, there is almost nothing about the applied context that seems to influence the design, nor the analysis of the results beyond the standard ML metrics.

---

> ### Author Response · Authors · 2021-11-23
> **Response to Reviewer 56Ru**
>
> First of all, we appreciate the valuable feedback from reviewer 56Ru and we are happy to use this chance to discuss further. Here we summarize the questions by reviewer 56Ru and try our best to answer each of them. We hope our discussion could address your concerns and increase the credibility of our work.
>
> Comment 3.1.It is not clear to what extent this is specific to surgery.
>
> Response:We agree with our proposed method can be generalized to other application domains that use high-dimensional,  complex,  and noisy data and value interpretability.   While this work was motivated by and evaluated based on real-world applications in surgeries, we have added discussions about the generality of the proposed approach in the abstract, introduction and conclusion.
>
> Comment 3.2.The result is Eq 6 and SVD is subsequently used in the estimation/replacement of TC. It would have been advisable to dig into this substation theoretically, a bit more.
>
> Response: We have rewritten this section in the manuscript according to your advice. In essence, our work is motivated by the previous literature that:  1) batch estimation of TC gave underestimations when using KL divergence, which deteriorates when dimensionality increases.  The empirical evidence were given in Appendix A6 and also in the Appendix A by Kim & Mnih (2018) and the theoretical discussion was given by Cheng et al. (2020). 2) The estimation gave inconsistent results for mean representation and the resampled estimation, as empirically demonstrated by Locatello et al. (2019). 3) The remedy strategies proposed by Chen et al. (2018) are not accurate for high dimensional latent space, as discussed by Cheng et al. (2020). To address this, we have shown that our proposed TC estimation is
> 1.  Simple and efficient, with no parameter tuning required.
> 2.  Bounded regardless of latent dimensionality.
> 3.  Minimized if and only if the latent space is disentangled.
> 4.  Bounded by the actual value if using all data, and the average of batch estimations forms the upper bound of the actual value. When NN converges, we are guaranteed with the minimization of the TC estimation using whole data.
>
> We have discussed about these properties in the Appendix A based on the Courant-Fischer min-max theorem. To demonstrate the non-triviality of the proposed work, we extended our empirical analysis on three datasets, and observed consistently better disentanglement in three datasets.
>
> Comment 3.3. How the (severe) missing rate in the delirium data was handled is not expressed.
>
> Response: In this paper, we focused on the downstream learning process and have used standard mean imputation to process the input data. Meanwhile, we acknowledge the recent advances of VAE-based imputation, and see the potential of combining VAE-based prediction with imputation. Further studies are ongoing.
>
> Comment 3.4. Instead of choosing a single standard for all baselines, it would have been preferable to under-stand their strengths and weaknesses at different hyper-parameters.
>
> Response: According reviewer’s advice, we have extended our study to train all VAE variants with different latent dimensionality, and tabulated their prediction performance, disentanglement, and reconstruction in Appendx A6-8.  The extensive results further validate our claim that sVAE is superior in prediction and disentanglement against all dimensionality.
>
> Comment 3.5. Although sVAE clearly outperforms many baselines, the degree to which it’s close to smooth-VAE
>
> Response: The superiority of sVAE lies in its capability to achieve both predictive performance and interpretability. In term of predictive performance, sVAE significantly outperforms all the base-line methods except smooth-VAE. The marginal improvement over smooth-VAE is not surprising because both sVAE and smooth-VAE are designed to improve predictive performance by integrating the predictive task in latent encoding.  However, unlike sVAE, smooth-VAE cannot disentangle the latent space for interpretability.  We have empirically compared sVAE and smooth-VAE in term of disentanglement.  Our results shows that sVAE outperforms smooth-VAE significantly in term of disentanglement, with 13\% $\sim$ 46\% improvement in UDR in different datasets (Table 4).  The comparisons between sVAE and smooth-VAE demonstrate that sVAE can effectively disentangle the latent space without incurring penalty in predictive performance.  Overall, in comparison to the SOTA models designed to disentangle the latent space, sVAE provides advantages in disentanglement while significantly outperforming them in predictive performance.  At the same time, sVAE disentangles the latent space while maintaining marginal performance advantage over existing methods optimized for predictive performance.   The unique combination of performance and disentanglement makes sVAE a particularly suitable approach for applications (e.g., healthcare) with stringent requirements in both accuracy and interpretability.

---

> > ### Author Response · Authors · 2021-11-23
> > **Response to Reviewer 56Ru - cont'd**
> >
> > Comment 3.6. The interpretability angle is interesting and appreciated, especially within the medical con-text. Not enough has been shown to warrant the mention of ‘trust’ in the conclusion, though.
> >
> > Response: We have removed the mention of "trust" from the conclusion as suggested. In the qualitative analysis, we want to demonstrate how disentangled latent encoding enhances interpretability and ”trustfulness” in two aspects:  the clustering of samples based on clinical phenotypes and the organization of samples according to existing clinical evidence.  The former aspect is demonstrated with the learning of intrinsic textual similarities, and the latter aspect is presented with the clinical evidence that delirium is linked to albumin levels, and when transforming feature ”albumin level” into the latent dimension, we see that it has been decomposed into two dimensions:  the prediction dimension representing the risk of getting delirium, and the other dimension representing the remaining information in ”albumin level” that is irrelevant to delirium prediction. While a helpful step in the direction, we agree that we have yet to demonstrate trust in this paper and hence have removed it from the conclusion.
> >
> > Comment 3.7. How does the applied context influence the design?
> >
> > Response:  Our work is driven by the clinical need of prediction performance and interpretability of a deep model given high-dimensional,  complex (and noisy) input data. While these challenges and characteristics are not unique to healthcare, the critical nature of surgeries and the nature of electronic perioperative data serve as a representative application that would benefit from sVAE. Furthermore, we show for the first time the use of VAE to help characterize patients, where the latent space is designed to not only explicitly express the prediction risk, but also uncover the phenotypes.  Our design is built on the fact that existing VAE models with disentanglement do not benefit downstream learning (Locatello et al., 2019) and the disentanglement  would  fail  when  latent  dimensionality  is  not  small,  which  is  the  truth  for  EHR data.  We revised our paper accordingly so that the two features of our model ( transformation function $g(z)$ of latent space and the new estimation method $TC(v)$) both theoretically and empirically demonstrate their necessarity in prediction and latent space disentanglement.
> >
> > Again, we thank the reviewer for their valuable feedback and we look forward to the follow-up discussions to address all concerns. Thank you and Happy Thanksgiving!

---

### Official Review · Reviewer_4Kio · 2021-11-01

**Correctness:** 4
**Technical Novelty And Significance:** 3
**Empirical Novelty And Significance:** Not applicable
**Recommendation:** 5
**Confidence:** 4

**Main Review:**

1) Regarding reproducibility, an experiment using a public dataset is more important. Thus it would be better to have MIMIC-III dataset results in the main text rather than one of the private datasets.

2) While the authors argue that this sVAE is specifically designed for high-dimensional and complex clinical data, there are many prediction tasks in various fields using such high-dimensional and complex, (and noisy) data. Thus the authors should at least discuss generalization of this approach.


**Summary Of The Paper:**

To improve accuracy of prediction and interpretability on a task using high dimensional and noisy data, the authors proposed surgical VAE (sVAE). sVAE is prediction-guided with explicit expression of predicted outcome in the latent representation, and it disentangles the latent space so that it can be more interpretable. sVAE was evaluated using two private datasets and one public dataset, and showed its efficacy on both classification tasks and regression tasks.


**Summary Of The Review:**

The authors introduced sVAE, a prediction-based, interpretable deep latent model designed for clinical prediction tasks. However, such tasks based on high-dimensional and complex data exist not only in surgery but also in a wide range of fields in the real world, thus generalization is important considering the backgrounds of ICML participants. If the authors aim to keep the scope within the clinical or surgical field, a more specific venue would be appropriate.

---

> ### Author Response · Authors · 2021-11-23
> **Response to Reviewer 4Kio**
>
> We appreciate the valuable feedback from reviewer 4Kio and we are happy to use this chance to discuss further. Here we summarize the questions by reviewer 4Kio and try our best to answer each of them.
>
> Comment 2.1.Regarding reproducibility, an experiment using a public dataset is more important.Thus it would be better to have MIMIC-III dataset results in the main text rather than one of the private datasets.
>
> Response:We have moved MIMIC-III to the main text.  As the observations are consistent across all datasets and models, this change does not affect our conclusions. To further show the generalizability, we have extended the analysis on MIMIC-III to demonstrate the consistent superiority of our proposed model in terms of prediction and disentanglement.
>
> Comment 2.2.While the authors argue that this sVAE is specifically designed for high-dimensional and  complex  clinical  data,  there  are  many  prediction  tasks  in  various  fields  using  such  high-dimensional and complex, (and noisy) data. Thus the authors should at least discuss generalization of this approach.
>
> Response:  We agree that our proposed method can be generalized to other application domains that use high-dimensional, complex, and noisy data and value interpretability. While this work was motivated by and evaluated based on real-world applications in surgeries, we have added discussions about the generality of the proposed approach in the abstract, introduction and conclusion.
>
> Again, we thank the reviewer for their valuable feedback and we look forward to the follow-up discussions to address all concerns. Thank you and Happy Thanksgiving!

---

### Official Review · Reviewer_CEnW · 2021-11-02

**Correctness:** 3
**Technical Novelty And Significance:** 2
**Empirical Novelty And Significance:** 3
**Recommendation:** 6
**Confidence:** 4

**Main Review:**

The proposed method combines the idea of disentangled latent variables and identifying meaning of representation for the prediction task. It also provides a new estimation for the correlation between latent variables, which also showed superiority in the conducted experiments. There is novelty in the methodology, however, the marginal improvement of the proposed methods over the previously published methods may not be sufficient to justify its superiority.
Besides that, I may have the following questions:
1. The authors also suggested a new measure of correlations between the latent variables and  to use that instead of TC. The new correlation metric was compared with TC in the delirium dataset. Was this consistently observed in other datasets?
2. There are two hyper parameters introduced in this method, but the authors did not mention the reasons for the choice of the hyper parameters. When we think about an end-to-end modelling process, the hyperparameter tuning and model selection process might also play a role in the evaluation of the method. Will the authors help to explain the choice of the hyperparameters?


**Summary Of The Paper:**

The paper proposed a modification of the VAE that was designed for handling ICU data for perioperative care. The new method specifically handles the entanglement of the latent variables in the traditional VAE, so it can provide interpretable representation of the data that also serves well the downstream prediction and regression tasks.
The authors conducted experiments on one publica dataset and two new real-world perioperative datasets. In all cases, the proposed methods are superior to the comparator methods.


**Summary Of The Review:**

Compared with the previously published methods, however, the new method did not show a significant superiority. However, it was a good application of VAE for an important clinical question.

---

> ### Author Response · Authors · 2021-11-23
> **Response to Reviewer CEnW**
>
> We appreciate the valuable feedback from reviewer CEnW and  we are happy to use this chance to discuss further. Here we summarize the questions by reviewer CEnW and try our best to answer each of them.
>
>
> Comment 1.1. The marginal improvement of the proposed methods over the previously published methods may not be sufficient to justify its superiority.
>
> Response: The superiority of sVAE lies in its capability to achieve both predictive performance and interpretability. In term of predictive performance, sVAE significantly outperforms all the baseline methods except smooth-VAE. The marginal improvement over smooth-VAE is not surprising because both sVAE and smooth-VAE are designed to improve predictive performance by integrating the predictive task in latent encoding. However, unlike sVAE, smooth-VAE cannot disentangle the latent space for interpretability. We have empirically compared sVAE and smooth-VAE in term of disentanglement. Our results shows that sVAE outperforms smooth-VAE significantly in term of disentanglement, with 13\% $\sim$ 46\% improvement in UDR in different datasets (Table 4). The comparisons between sVAE and smooth-VAE demonstrate sVAE can effectively disentangle the latent space without incurring penalty in predictive performance. Overall, in comparison to the state of art models designed to disentangle the latent space, sVAE provides advantages in disentanglement while significantly outperforming them in perdictive performance. At the same time, sVAE disentangles the latent space while maintaining marginal performance advantage over existing methods optimized for predictive performance. The unique combination of performance and disentanglement makes sVAE a particularly suitable approach for applications (e.g., heathcare) with stringent requirements in both accuracy and interpretability.
>
> Comment 1.2. The authors also suggested a new measure of correlations between the latent variables and to use that instead of TC. The new correlation metric was compared with TC in the delirium dataset. Was this consistently observed in other datasets?
>
> Response: Yes. The better performance was consistently observed in all datasets for all the metrics we have used. To show generalizability and reproducibility, we have repeated the same experiments on the open MIMIC-III dataset and uploaded the results to Appendix (Table A3, A4 and A8).
>
>
> Comment 1.3. There are two hyper parameters introduced in this method, but the authors did not mention the reasons for the choice of the hyper parameters. When we think about an end-to-end modeling process, the hyperparameter tuning and model selection process might also play a role in the evaluation of the method. Will the authors help to explain the choice of the hyperparameters?
>
> Response: We chose $\beta=10$ based on previous literature on disentanglement ( Chen et al. (2018) ), and $w=1$ by default. In response to this comment, we empirically explored a range of hyperparameter settings for $\beta$ and $w$, and reported their prediction performance in Appendix Table A9. As can be seen, the model is robustly better than all baseline models with any choice of hyper-parameters. In practice, the hyperparameters may be selected through grid search.
>
> Again, we thank the reviewer for their valuable feedback and we look forward to the follow-up discussions to address all concerns. Thank you and Happy Thanksgiving!

---

### Decision · Program_Chairs · 2022-01-20

**Decision:**

Reject

**Comment:**

The reviews are of good quality. The responses by the authors are commendable, but ICLR is selective and reviewers still believe that the research would be better as two separate papers: one about the problem and solution from an ML perspective, and the other about the application to surgery. Papers that provide a new method in the context of a single application domain run the risk of making a contribution to neither, and of being evaluated by reviewers who are not experts in both.